# Styrene Monomer Levels in Polystyrene-Packed Dairy Products from the Market versus Simulated Migration Testing

**DOI:** 10.3390/foods12132609

**Published:** 2023-07-06

**Authors:** Valeria Guazzotti, Veronika Hendrich, Anita Gruner, Angela Störmer, Frank Welle

**Affiliations:** Fraunhofer Institute for Process Engineering and Packaging, IVV, Giggenhauser Straße 35, 85354 Freising, Germany

**Keywords:** styrene, migration, polystyrene, food packaging, compliance testing

## Abstract

In view of the fact that a specific migration limit (SML) is to be established in the near future for styrene monomer in plastic food contact materials (FCMs), data on the dietary exposure of the European population, as well as sensitive and reliable analytical methodologies to implement compliance testing, are needed. The properties of the substance styrene as well as those of styrenic polymers pose challenges for analysts and their design of experimental migration tests. The aim of this study was to assess the level of styrene in polystyrene (PS)-packed dairy products from supermarkets and compare these values with the results from simulated migration testing. In addition to the conventional food simulant and test conditions described in Regulation (EU) No 10/2011 for refrigerated dairy products (50% ethanol for 10 days at 40 °C), milder simulants and test conditions (10% ethanol and 20% ethanol for 10 days at 40 °C and 20 °C) were investigated. Styrene levels in the investigated foods ranged from 2.8 µg/kg to 22.4 µg/kg. The use of 50% ethanol causes interactions with PS (swelling) that do not occur with dairy products and leads to highly exaggerated migration results. In contrast, testing PS for 10 days at 40 °C with 10% and 20% ethanol leads to higher styrene migration levels than found in real food, which are still conservative but far less extreme. Testing PS for 10 days at 20 °C leads to styrene migration levels that are more comparable to, but still overestimate, those found in real food products stored under refrigerated conditions.

## 1. Introduction

Food contact materials (FCMs) placed on the European market must comply with Regulation (EC) No 1935/2004 [1], which sets out the general principles of safety and inertness for all materials and articles intended to come into contact with food. In addition to the general legislation, FCMs made of plastic must comply with Commission Regulation (EU) No 10/2011 [2], which sets out rules on the composition of plastic FCMs and establishes a Union List of substances that are permitted for use in their manufacture. Regulation (EU) No 10/2011 also specifies restrictions on the use of these substances and sets out rules to determine the compliance of plastic materials and articles. An important mechanism to ensure the safety of plastic materials is the use of migration limits. These limits specify the respective maximum amount of substance allowed to migrate from the FCM to food. For the substances on the Union list, the Regulation sets out ‘Specific Migration Limits’ (SMLs). These are established by the European Food Safety Authority (EFSA) based on available toxicity data. The Regulation also sets out detailed rules on migration testing. Although migration testing in food prevails, migration is usually tested using ‘simulants’. These simulants are used to represent a defined food category, e.g., 50% aqueous ethanol is assigned for dairy foods. The migration testing is done under standardized time/temperature conditions, as defined for a specific food use, and covers the maximum shelf-life of packaged food. For example, testing for 10 days at 40 °C covers all storage times in refrigerated and frozen conditions, including hot fill. The safety of FCMs is tested by the business operators placing them on the market and by the competent authorities of the Member States during official controls.

Among the several types of plastics, styrene polymers are extensively used in many food contact applications. They are a family of amorphous thermoplastic polymers that use styrene as the key building block (monomer). Polystyrene (PS) is the first representative of styrene polymers, and thanks to its ease of processing, mechanical strength, lightweight, water resistance, and relatively low production cost, PS is recognized as a versatile material for the food packaging sector [3]. The main food contact application for PS is refrigerated dairy packaging [4], with the ‘yogurt pot’ being the top application [5]. Other important food contact applications of PS are rigid containers and foamed trays for packaging fresh meat, fish, poultry, or cheese. For most of these applications, refrigerated storage and short-term contact are foreseen. PS packaging is also used for ‘single serving packaging’, a type of packaging used to contain one single serving of a food product that can be refrigerated, e.g., fresh cheese, or stored at room temperature, e.g., jam, spreadable chocolate, sterilized coffee creamer, or milk. PS has often been used for “disposable” applications (e.g., dishware, cutlery, beverage straws, and cups/containers for take away food). However, Directive (EU) 2019/904 [6] on the reduction of the impact of certain plastic products on the environment greatly impacts the use of PS in disposable/single-use applications through, among others, a ban on certain single-use plastic products.

Styrene monomer is the building block from which PS polymer is produced. During manufacture, a small amount of the monomer styrene remains unreacted in the polymer and, as a result, is available in the final food contact article to migrate into the foodstuff. The migration of styrene from PS packaging into various foods and simulants has been extensively studied in recent decades. Styrene migration strongly depends on its residual level content in the polymer, the type of polymer, the fat content of the food, as well as the storage time and temperature [7,8,9,10,11,12,13,14,15].

According to the current European Food Packaging Legislation, styrene is an authorized monomer, listed without a specific restriction (SML). The only restriction in the EU legislation regarding the migration levels of styrene in food is set by Art. 3 of the Framework Regulation (EC) No 1935/2004, which prohibits the transfer of constituents from materials that cause deterioration of the organoleptic characteristics of the food. Odor and taste thresholds for styrene are reported to range from 4 to 6000 µg/kg, depending on the type of food [16,17,18]. Styrene was identified as the cause of 15% of all off-flavors caused by packaging material [19]. To prevent any styrene taint issues, the industry has taken measures to decrease the residual styrene in PS for food contact applications to below 500 mg/kg [20].

In 2019, the International Agency for Research on Cancer (IARC) [21] classified styrene as “probably carcinogenic to humans,” and according to a first assessment made by EFSA in 2020, a further review of the available data is required before concluding on the safety of styrene in FCM [22]. Recently, the European Commission’s Directorate-General for Health and Food Safety (DG SANTE) presented the intention to set a precautionary SML of 40 µg/kg for styrene [23] and proposed that testing would be in food, not simulants.

Testing using food simulants is an essential pillar of the Plastics Regulation (EC) No 10/2011 and therefore essential for the whole value chain to demonstrate compliance of FCMs placed on the European market. However, the properties of the substance styrene as well as those of PS and other styrenic polymers or copolymers create significant challenges for analysts and the experimental design of migration tests. On the one hand, migration testing employing simply water may lead to an underestimation of the migration processes occurring from PS FCMs into real food like fatty and dairy products [24]. On the other hand, it has been recognized that certain food simulants, such as high concentrations of ethanol/water mixtures, n-heptane, or isooctane, strongly interact with the polymer matrix, leading to over-estimation of the migration and excessively high diffusion coefficients as a result of swelling (liquid uptake) and stress cracking [25,26,27]. In previous studies [26,27,28], we investigated the occurrence/correlation between swelling and styrene migration from styrene polymers into food simulants and real foods using mathematical modeling, showing that the strong swelling of certain PS polymers by certain simulants (especially at elevated temperature conditions) overestimates the styrene migration compared to the migration into foodstuffs under real use conditions. Additionally, in one of our recent studies [29], yogurt and cream products packed in PS pots taken from the Italian and German markets in 2021 were investigated with regard to their migrated styrene levels. We found styrene concentrations in these refrigerated dairy products at their shelf-life ranging from 5 to 30 µg/kg (taking into account the real ratio of the PS surface area of the package to the quantity of the food packed for each pot), corresponding to 3 to 20 µg/kg (applying the EU cube model: 6 dm²/kg). These results are in close agreement with published data on styrene concentration in yogurt desserts and cream packed in PS containers, which were sampled from the Greek market in 2020 [30].

Actual migration of styrene into refrigerated dairy food packed in PS appeared to be much lower than that measured within laboratory simulations under standardized conditions according to Regulation (EU) No 10/2011. This raises the question of whether compliance testing should be adapted for PS polymers to take into consideration the correlation between migration in food simulants and in real food. The use of food as opposed to simulants in the migration tests can be considered the ideal approach for the evaluation of consumer exposure to styrene. However, it has to be recognized that, in practice, the use of simulants is more practical and necessary for routine compliance testing. 

The aim of this study was to assess the level of styrene in different refrigerated dairy products taken from the supermarket and compare these values with the results from migration tests in food simulants. In order to resolve the issue of highly exaggerated migration results obtained using 50% ethanol for 10 days at 40 °C, alternative/milder simulants and test conditions (10% ethanol and 20% ethanol for 10 days at 40 °C as well as for 10 days at 20 °C) were investigated for their suitability to demonstrate compliance of PS packaging for refrigerated dairy products. 

## 2. Materials and Methods

### 2.1. Collection of Dairy Food Products from the Market

Refrigerated dairy products, including yogurt, cream, milk dessert, milk and rice pudding, quark, fresh and cottage cheese, and dessert for kids, all packed in pots, cups, or mini-tray portions, were purchased during the year 2022 from different German supermarkets.

Products of different brands were chosen, looking at the bottom of the pots for their labeling for the recycling code for PS of 6. For some products, the recycling code was not reported, in which case analysis of styrene monomer and oligomer residual levels in the packaging was used to identify them as PS. The main characteristics of the packaged food samples collected from the market for the present study (with regard to fat and net content as well as surface/volume ratios between the contact surface areas of the packaging and the mass of food) are reported in the Results section.

### 2.2. Determination of the Residual Level of Styrene Monomer in the PS Packaging Materials 

The concentrations of the styrene monomer were determined in the PS packaging (after emptying out the food at its best before date (BBD) and gently washing with water) by extraction with the solvent acetone (Chemsolute, Th.Geyer GmbH, Renningen, Germany) and subsequent analysis by gas chromatography with flame ionization detection (GC-FID) as described in one of our previous publications [27]. In brief, 1.0 g of the material was extracted with 10 mL of acetone and stored at 60 °C for 3 days. Triplicate experiments were performed. The extracts were analyzed by gas chromatography with flame ionization detection (GC-FID). The Agilent 6890 GC/FID system (Agilent Technologies, Santa Clara, CA, USA) was used with the following conditions: column: DB 1 (length 20 m, inner diameter 0.18 mm, film thickness 0.18 µm); GC temperature program: 50 °C (2 min), followed by heating at 10 °C/min to 340 °C (15 min); pre-pressure: 50 kPa hydrogen; split: 10 mL/min. The injection volume was 5 µL with a split ratio of 1:20. An internal standard solution of di-(tert-butyl)-hydroxyanisol (BHA) (Fluka, Seelze, Germany) and Tinuvin 234 (Ciba, Basel, Switzerland) was used for all GC analyses to check the stability of the retention times and the performance of the GC analysis. The quantification of monomer styrene was achieved through external calibration using a reference standard (Tokyo Chemical Industry Co., Ltd., Tokyo, Japan). To avoid interference, only parts of the packaging without labels or printing (if present) were analyzed. 

### 2.3. Determination of the Concentration of Styrene Monomer in the Dairy Food Products from the Market by Purge and Trap (P&T) Gas Chromatography (GC)

After buying food in the supermarket, the food products were refrigerated at 5 °C. At the best before date (BBD) of each sample (as reported on each packaging from the producer), the samples were opened, homogenized, the content entirely transferred to PP cups, and immediately frozen until analysis. 

Styrene was determined by purge and trap gas chromatography coupled with both mass spectrometry and a flame ionization detector (P&T/GC/MS + FID) (Thermo Scientific Trace GC, with VSP Autosampler, IMT, Moosbach, Germany) according to our recently published method [29], with some modifications. In brief, 1 g of each food (thawed and mixed) was slurried with 5 mL of freshly boiled and cooled-down water (HPLC grade) and spiked with an internal standard solution of styrene d-8 (CAS 19361-62-7) (Sigma-Aldrich, Saint Louis, MO, USA). The headspace of the sample vials was heated to 40 °C and purged with helium for 20 min at a flow rate of 20 mL/min. The column used was a Restek RXI 624 MS Sil with 60 m length, 0.32 mm inner diameter, and 1.8 µm film. The temperature program was 40 °C for 6 min, then heating with 5 °C/min to 90 °C, then heating again with 10 °C/min to 260 °C for 15 min. The mass spectrometer was set in full scan mode (35–350 amu) at 0.2 scans/s. A splitter separated the eluting substances 1:1 to MS and FID. Quantification was obtained using the MS response. The monitored ions are *m*/*z* 104 for styrene and *m*/*z* 112 for styrene d-8. All measurements of the samples were performed in triplicate.

The P&T/GC/MS method was evaluated in terms of linearity, limit of detection (LoD), limit of quantification (LoQ), accuracy, and precision. External calibration using a matrix blank sample (3.5% fat yogurt, plain, packed in PP) spiked with styrene and styrene d-8 as internal standards was performed. Six concentration levels of styrene, diluted from a stock solution prepared in dimethylacetamide, were analyzed. The linearity was evaluated by the linear regression coefficient (R^2^) of the calibration curve constructed based on the ratio of the analyte to the internal standard detector response. The LoD and the LoQ were evaluated based on the chromatographic signal-to-noise ratio S/N obtained from the chromatograms of the lowest calibration level. The LoD was calculated as three times the S/N, while the LoQ was calculated as 10 times the S/N. Short-term repeatability was established by analyzing six replicates of the matrix blank sample spiked with a styrene standard (26.4 µg/kg) during the day, while for intermediate precision, two replicates of the spiked blank sample were analyzed on eleven consecutive days. All the dairy food products were fortified with a styrene standard of 21 µg/kg. The relative recovery for each product was calculated as the amount of styrene found in the sample after fortification subtracted by the amount found in the neat sample divided by the known amount added, expressed as a percentage. Additionally, a standard addition was performed on selected products (Table 1 samples: 8a, 8b, 19a, and 19b) by spiking them at three concentration levels (depending on the initial value of styrene found in each product).

### 2.4. Determination of the Specific Migration of Styrene Monomer into Food Simulants by Headspace GC-FID

The determination of the specific migration of styrene monomer into food simulants was tested using two PS materials: thermoformed PS yogurt pots and extruded high-impact polystyrene (HIPS) sheets. The PS pots were purchased from a supermarket (see description in Table 1, sample 29; in total, 10 pots were purchased with the same lot number). These had a thickness of approx. 130 µm and an area weight of approx. 2.02 g/dm^2^. The food content (plain yogurt) was emptied (approx. 22 days before expiration), the pots were washed gently, and they were cut into strips for the migration experiments. The extruded sheets were made of HIPS and were provided by a polystyrene manufacturer. The HIPS sheets had a thickness between 290–360 µm and an area weight of approx. 3.17 g/dm^2^. The residual content of styrene in the PS pot and HIPS sheets was determined by solvent extraction (acetone) and GC-FID analysis as described in Section 2.2.

The tested food simulants were 50% ethanol, 20% ethanol, and 10% ethanol. The applied test conditions were for 10 days at 40 °C and 10 days at 20 °C. Migration contacts were performed according to European Standard EN 13130-1:2004-08 [31]. Test specimens with a contact surface of 0.5 dm^2^ (total surface excluding the area of the cut edges) were cut from the PS pot and HIPS sheets samples and placed in stainless steel migration cells containing 80 mL of each food simulant in such a way that the cells were entirely filled with simulants, minimizing the remaining headspace. The cells were then placed in thermostatically controlled ovens. Each test was carried out in triplicate. Recovery experiments were performed by spiking a styrene standard (56.6 µg/L) in the simulants under the same conditions used for the migration contact of the samples. 

Simultaneously to the migration experiments, the weight increase of the test specimen after contact with simulants (uptake of liquid corresponding to the swelling extent) was assessed by weighing them immediately before and after migration contact. 

The specific migration of styrene in simulants was measured by headspace GC-FID (Agilent Technologies, Santa Clara, CA, USA). The calibration was performed in the respective simulant using m-xylene (CAS: 108-38-3) (Sigma-Aldrich, Saint Louis, MO, USA) as the internal standard. The 50% ethanol migrates were diluted to 20% ethanol. The samples were equilibrated at 80 °C for 1 h, and then the gas phase was transferred into a GC (needle temperature 110 °C, transfer line temperature 120 °C). Separation was carried out on a ZB 624 column (60 m, 0.32 mm, 1.8 µm) at an initial temperature of 60 °C (2 min), heated 10 °C/min to 250 °C (5 min). Quantification was carried out by external calibration in 20% ethanol. All measurements were performed in triplicate.

## 3. Results

The method was validated in-house. The squares of the correlation coefficients (R^2^) were higher than 0.998 within the range of interest (2.1–52.5 µg/kg). The method detection limit (LoD) in yogurt was 0.4 µg/kg, and the limit of quantification (LoQ) was 1.4 µg/kg. The accuracy for styrene in yogurt (mean recovery of plain yogurt spiked with styrene and styrene d-8) was 105.2% with an inter-day relative standard deviation of 5.4%. The relative recovery in each food product ranged from 95.7% to 112.5%; the standard addition applied to selected products (Table 1 samples: 8a, 8b, 19a, and 19b.) confirmed the accuracy of the method.

Table 1 summarizes the migration levels quantified in food (refrigerated dairy products from the supermarket) as well as the content of styrene monomer (*Cp,0*) determined in each of the respective packaging polymers. In the same table, fat content (absolute) of the products and the surface/volume (S/V) ratios calculated by dividing the contact surface areas of the packaging by the mass of food are reported. Migration values are given in µg/pot, µg/dm^2^ and µg/kg (food), calculated considering the real S/V ratio for each product and applying the EU cube model: 6 dm^2^/kg. The migration of styrene was measured at the best-before date (BBD), as reported on each product by the producer. For some products, additional determinations were made before and/or after BBD (see values in italics in the table).

**Table 1 foods-12-02609-t001:** Residual level in the packaging and migration of styrene in refrigerated dairy products (from the supermarket) at BBD (*in Italics before/after BBD*).

Code	Food Description	Fat Content	S/V	*Cp,0*	Styrene Migration in Food			
		[%]	[dm^2^/kg]	[µg/g]	[µg/pot]	[µg/dm^2^]	[µg/kg] ^1^	[µg/6 dm^2^] ^2^
**Dairy products packed in PS pots/cups**								
1	Rice pudding	2.6	8.2	251 ± 7	0.96 ± 0.08	0.59 ± 0.05	**4.8**	**3.5**
2	Low-fat fermented milk drink	1.5	6.4	317 ± 4	4.26 ± 0.02	1.34 ± 0.01	**8.5**	**8.0**
3a	Buttermilk. Brand-1	1.0	6.4	310 ± 6	2.75 ± 0.02	0.86 ± 0.01	**5.5**	**5.2**
3b	Buttermilk. Brand-2	1.0	6.6	381± 4	4.91 ± 0.14	1.49 ± 0.04	**9.8**	**8.9**
4a	Low-fat curd cheese. Brand-4	0.4	8.6	348 ± 4	1.12 ± 0.02	0.52 ± 0.01	**4.5**	**3.1**
4b	Semi-fat curd cheese. Brand-4	4.5	8.6	362 ± 2	1.99 ± 0.11	0.93 ± 0.05	**8.0**	**5.6**
4c	Fat-cream curd cheese. Brand-4	14	8.6	305 ± 5	3.35 ± 0.11	1.56 ± 0.05	**13.4**	**9.4**
5	Low-fat yogurt for kids	1.5	12.2	246 ± 5	0.45 ± 0.03	0.47 ± 0.03	**5.7**	**2.8**
10a.	Dessert for kinder: fresh cheese with fruit (apricot) preparation	2.5	12.6	310 ± 8	0.46 ± 0.02	0.73 ± 0.04	**9.3**	**4.4**
10b.	Dessert for kinder: fresh cheese with fruit (banana) preparation	2.5	12.6		0.42 ± 0.06	0.67 ± 0.09	**8.4**	**4.0**
11	Skyr (made of 70% low-fat quark)	0.2	6.2	289 ± 4	2.10 ± 0.05	0.75 ± 0.02	**4.7**	**4.5**
12	Cottage cheese	4.6	8.3	368 ± 6	1.06 ± 0.04	0.64 ± 0.02	**5.3**	**3.9**
13	Cream fine for cooking.	15	8.2	244 ± 1	2.77 ± 0.06	1.70 ± 0.04	**13.8**	**10.2**
15	Fruit (mix red fruits) yogurt with 25% fruit content	3.8	10.0	326 ± 5	1.63 ± 0.08	1.09 ± 0.05	**10.9**	**6.5**
16	Fruit (strawberry) yogurt (with 9% fruit content)	3.8	9.9	332 ± 7	1.20 ± 0.01	0.81 ± 0.01	**8.0**	**4.9**
17 ref.	Fruit yogurt in PP ^3^	1.5	-	-	<LoD	<LoD	**<LoD**	**<LoD**
*29a*	*Stirred mild yogurt. 23 d before BBD*	*3.5*	*8.6*	*310 ± 5*	*0.78 ± 0.05*	*0.45 ± 0.03*	*3.9*	** *2.7* **
29b	at BBD				1.67 ± 0.08	0.98 ± 0.05	8.3	**5.9**
*29c*	*31 d after BBD*				*1.85 ± 0.08*	*1.08 ± 0.05*	*9.3*	** *6.5* **
**Dairy products packed in PS single serving portions**								
*8a*	*Fresh cheese double cream approx. 20 d before BBD*	*23*	*20.0*	*274 ± 2*	*1.05 ± 0.03*	*2.63 ± 0 07*	** *52.6* **	** *15.8* **
8b	at BBD				1.49 ± 0.04	3.72 ± 0.11	**74.5**	**22.4**
19a	Fresh cheese double cream at BBD	21	17	361 ± 4	0.71 ± 0.02	2.38 ± 0.09	42.9	**14.3**
*19b*	*approx. 30 d after BBD*				*0.82 ± 0.04*	*2.74 ± 0.14*	*49.4*	** *16.5* **

^1^ Calculated considering the real S/V ratio (dm^2^/kg) for each product. ^2^ Calculated applying the EU cube model: 6 dm^2^/kg. ^3^ Product used as reference product (matrix blank sample), packed in polypropylene (PP). LoD: Limit of Detection = 0.4 µg/kg.

Table 2 summarizes the residual content of styrene monomer (*Cp,0*) determined in the samples PS pots (code 29) and HIPS sheets, as well as the specific and relative migration into food simulants at different testing conditions. The relative migration states the percentage of substance that migrates from the polymer to the food simulant and was calculated by dividing the mass of styrene in the food simulant by the initial mass determined in the sample material. The weight increase determined for the test specimen after migration contact is also reported.

## 4. Discussion

### 4.1. Styrene Levels in Foods Packed in Polystyrene (Refrigerated Dairy Products from the Supermarket)

In this study, a total of eighteen refrigerated dairy products packed in rigid PS from different brands were investigated. Sixteen of these products were packed in PS pots/cups and included yogurts, puddings, desserts for children, and creams for cooking. The absolute fat content of these products ranged from 0.2% to 15%, and the surface/volume (S/V) ratio (calculated by dividing the contact surface areas of the packaging by the mass of food) ranged from 6.2 dm^2^/kg to 12.6 dm^2^/kg. The remaining two products investigated were PS single-serving portions of fresh cheese with an absolute fat content of 21% and 23% and an S/V of 17 dm^2^/kg and 20 dm^2^/kg, respectively. 

The residual content of styrene monomer in the rigid PS packaging of the investigated samples ranged from 240 to 380 µg/g polymer. These results are consistent with recently reported values for PS packaging used for refrigerated dairy products [29]. 

The styrene migration measured at BBD in the samples packed in PS pots/cups ranged from 4.5 µg/kg to 13.8 µg/kg (ppb, calculated considering the real S/V ratio for each product). The lowest migration values (4.5 µg/kg and 4.7 µg/kg) were measured for the products with lower fat content: sample 4a with 0.4% fat and 11 with 0.2% fat. The higher styrene migration values (13.4 µg/kg and 13.8 µg/kg) were measured in the higher fat products, respectively, sample 4c with 14% fat and sample 13 with 15% fat. The effect of fat content on the styrene migration levels can also be clearly seen in the results obtained for samples 4a, 4b, and 4c from the same brand, only differing in regard to their fat content. For these products, the styrene migration levels correlated positively with the fat content of the products. 

To avoid matrix interferences or styrene contamination from other origins than PS packaging, mostly plain dairy products (not containing flavors/spices/fruit/chocolate or other preparations) were chosen in this study. Four samples were analyzed that were not plain: two fruit yogurts (samples 15 and 16) and two fruit milk desserts for children (samples 10a and 10b). Fruits might contain styrene as a flavoring component [32], so a similar food product not packed in PS (i.e., a strawberry yogurt packed in polypropylene, sample 17 ref.) was analyzed. Styrene was not detectable in the reference sample (with a detection limit of 1.4 µg/kg).

As regards the two samples of single-serving portions of fresh cheese, the styrene migration levels at BBD were 74.5 µg/kg (sample 8b) and 42.9 µg/kg (sample 19a), calculated considering the real S/V ratio for each product. Therefore, the styrene migration is significantly higher compared to the other refrigerated dairy products packed in pots/cups. Possible explanations for this finding are the higher S/V of the single-serving portions compared to the ones in pots/cups, the high fat content of the fresh cheese, and the time of storage (product shelf life from production). In fact, according to the information provided by some dairy producers, most yogurts and milk desserts have a shelf life of approx. 40–50 days at refrigerated temperatures (8 °C), while refrigerated cream cheese spread can, according to information retrieved from the internet, have a shelf life of up to 7 months. However, the exact shelf life for each of the products investigated in the present study is unknown.

The two single-serving portions of fresh cheese and one yogurt product (sample 29) were additionally analyzed before and/or after BBD. The single portions were multipacks, and so individual samples could be analyzed at different time points, while for sample 29, three yogurt pots were simultaneously bought in the supermarket (they reported the same production lot number) and opened at different time points for analysis. The anticipated effect of increasing storage time on the migration levels of styrene was confirmed. Approximately 20 days before BBD, the styrene concentration in the single-serving portions of fresh cheese was 20 µg/kg lower than at BBD, and 30 days after BBD, it was only 7 µg/kg higher. Regarding the yogurt pot, the styrene concentration at 20 days before BBD was half of that at BBD. Surprisingly, 30 days after BBD, migration was barely increased (only 1 µg/kg higher). 

Applying the EU cube model (6 dm^2^/kg), the migration levels of styrene in the investigated samples (refrigerated dairy products) at BBD ranged from 2.8 µg/kg to 22.4 µg/kg. This conversion is applied to check the compliance with the relevant migration limits of plastic food packages that have a volume lower than 500 milliliters or grams or more than 10 L (except for those intended to come into contact with food intended for infants and young children) according to Regulation (EU) No 10/2011. The higher levels were found in the single-serving portions of fresh cheese (14.3 µg/kg and 22.4 µg/kg), while they remain below or at 10 µg/kg in all the other investigated products packed in PS pots/cups. All will be in compliance with the proposed SML for styrene of 40 µg/kg.

These results are in close agreement with recent published data on styrene concentration in refrigerated yogurt desserts and cream packed in PS containers, which were taken from the Italian, German, and Greek markets [29,30], where levels were found to range from <1 to 54 µg/kg (applying the EU cube model).

### 4.2. Styrene Migration into Food Simulants (10%, 20%, and 50% Ethanol)

According to the EU Plastic Regulation (EC) No 10/2011, the prescribed food simulant for dairy products is 50% aqueous ethanol. Additionally, 10 days at 40 °C are the time and temperature contact conditions to be used for testing specific migration, which covers all storage times under refrigerated and frozen conditions, including hot fill (e.g., yogurt and milk desserts). 

In the present study, migration of styrene monomer from thermoformed PS pots (yogurt pots taken from the supermarket, code-29) and extruded HIPS sheets (provided by a polystyrene manufacturer) was determined at testing conditions of 10 days at 40 °C and 10 days at 20 °C using the food simulants 10%, 20%, and 50% ethanol. The aim of the study was to investigate the relationship between the migration results in food simulants (official and alternative) and styrene levels measured in the food as sampled from the market (such as yogurt and dairy products).

Initially, the residual content of styrene in the PS pot and sheets was determined by solvent extraction (acetone) and GC-FID analysis. The measured levels in the two sample materials were 310 ± 5 µg/g and 392 ± 8 µg/g, respectively, which is comparable with the overall mean value of 300 µg/g recently found for yogurt pots made of PS [29].

Migration contacts were performed by total immersion of the test specimen according to European Standard EN 13130-1:2004-08 [31]. Due to the volatility of styrene and its slight solubility in 10% and 20% ethanol, special attention was paid to its handling in the laboratory to prevent losses during the migration experiment. The recovery tests under migration test conditions were in all cases above 72%.

Migration of styrene from the HIPS sheets was found to be higher than that from the PS pot, which is not completely explained by the different residual contents of styrene in polymers but might additionally be due to the different composition of the materials. The pots taken from the supermarket, code-29, were most likely composed of a 50:50 GPPS and HIPS blend (typical composition of materials used for this application according to manufacturers), while the sheets were composed of 100% HIPS, which composition is not typically used for the application as yogurt pots but represents a worst-case situation, with the diffusion coefficients from HIPS being higher than from GPPS [28]). Another factor that can influence the higher migration found from the HIPS sheets could be the higher thickness of the sheets (approx. 300 µm) compared to that of the pots (approx. 130 µm) and the consequent greater “edge effect”, which is a known parameter, especially for some polymers like PS, influencing the migration extent when testing by total immersion [33].

For both the PS pots and the HIPS sheets, the migration extent of styrene depends on the ethanol content of the food simulant and on the applied temperature. It ranged between 10.7–25.5 µg/kg in 10% ethanol and between 27.3–68.5 µg/kg in 50% ethanol tested at 20 °C, and between 52.6–75.6 µg/kg and 166.5–353.1 µg/kg in the same simulants tested at 40 °C.

The relative migration of styrene (calculated by dividing the mass of styrene found in the food simulant by the initial mass determined in the sample material) also depends on the ethanol content of the food simulants. It ranges from approx. 0.3% in 10% ethanol to approx. 0.9% in 50% ethanol tested at 20 °C and from 1.1% to 4.7% at 40 °C. The relative migration of styrene from PS packaging in dairy products was reported in our previous publication [29] as ranging from 0.04% to 0.29%. This is consistent with the relative migration found in this study. Recently, Naziruddin et al. [34] reported relative migration values of styrene from HIPS pots in yogurt equivalent to 83–96%. However, such data appear to be questionable since the authors report initial styrene content in the respective HIPS pots equivalent to 90–122 ng/g (ppb: part per billion), which is approx. a factor of 1500 lower than the typical styrene residual level determined in our and other studies [35,36,37,38]. Such low levels are evidently unrealistic since normal residual styrene monomer levels in food contact styrenic polymers have been reported as being in the ppm range (from 20 to 1400 ppm, typically around 300 ppm) since the 80’s [35,36,37,38].

Along with the migration experiments, the swelling or plasticization effect (uptake of liquid, interaction between the food simulant and the plastic) after contact of the PS pots and HIPS sheets with the ethanolic simulants was assessed by gravimetry. The weight increase of the two sample materials was found to be dependent on the temperature of exposure as well as the type of simulant. It ranged from approx. 0.2% in 10% ethanol to approx. 1.2% in 50% ethanol tested at 20 °C and from 0.4% to 1.5% at 40 °C. The weight increase of the test specimen after contact with 50% ethanol was, in all cases, greater than 1%, indicating a stronger interaction of this food simulant with PS compared to the effect caused by 10% and 20% ethanol. 

### 4.3. Styrene Concentrations in Dairy Products Packaged in Polystyrene vs. Simulated Migration Testing

In the present study, styrene levels in refrigerated dairy products packaged in PS were compared with styrene levels in food simulants obtained under simulated migration testing. For the analysis of foodstuffs, a P&T/GC/MS method was used, whereas food simulants were analyzed by headspace GC-FID.

The PS yogurt pots (corresponding to samples 29a-b-c) were purchased from the supermarket and emptied and washed before being used for migration experiments with simulants. Styrene migration levels measured in yogurt (stirred, 3.5% fat content) were 0.98 µg/dm^2^ equivalent to 5.9 µg/kg (calculated applying the EU cube model: 6 dm^2^/kg) at BBD compared to 6.5 µg/kg 31 days after BBD. The present study shows that there is higher, or significantly higher, styrene migration from PS pots into ethanol-water food simulants (10% ethanol: simulant A, 20% ethanol: simulant C, and 50% ethanol: simulant D1, according to Regulation (EC) N. 10/2011) under conditions of 10 days at 40 °C compared with migration into food (yogurt) stored under refrigerated conditions. Styrene migration from the same PS pots into 50%, 20%, and 10% ethanol under the same conditions was 166.5 µg/kg, 75.6 µg/kg, and 52.6 µg/kg respectively. All these results obtained in food simulants would exceed the foreseen precautionary SML for styrene of 40 µg/kg [22]. However, this limit would not be exceeded in the yogurt packed in the same PS pot and stored until BBD under refrigerated conditions. Testing the PS pots after 10 days at 20 °C with the food simulants A, C, and D1 leads to more comparable (or slightly higher, depending on the type of simulant) styrene migration levels as found in yogurt (ranging from 10.7 µg/kg in 10% ethanol to 27.3 µg/kg in 50% ethanol), which would also be below the limit of 40 µg/kg foreseen by the EU Commission. 

The styrene migration levels measured in the food simulants A, C, and D1 from the extruded 100% HIPS sheets (provided by a PS manufacturer) were higher than those obtained from the PS pots (GPPS/HIPS blend) under comparable conditions, reaching 351.3 µg/kg in 50% ethanol at the condition of 10 days at 40 °C. This value would be approx. nine times higher than the precautionary SML for styrene of 40 µg/kg. However, it must be recognized that no data is available on migration from GPPS/HIPS blend sheets to real food, and 100% HIPS is a higher diffusivity polymer [28] not typically used to make yogurt pots. Notwithstanding this, the data indicates that styrene migration may be influenced by the different PS composition/blend. Indeed, HIPS is often blended with GPPS at the stage of production of the thermoforming sheet, the ratios being selected to achieve the required balance of physical properties for the different forms of packaging and the conversion process. Containers may also be produced in form-fill-seal packaging machines (where the completely packaged goods are produced in-line, starting from polymer granules, extrusion, thermoforming, filling, and sealing) or delivered in preformed packaging from packaging materials suppliers or converters. Reflecting this complexity, it is therefore not surprising that if manufacturers of PS FCMs do not have the opportunity to get the final thermoformed PS pots, they would likely test their own extruded polymer sheets for compliance. Irrespective of whether they test sheets or pots, if they used food simulant D1 in accordance with the food simulant assigned for dairy products by Regulation (EU) No 10/2011, there would be a considerable risk of obtaining an unrealistically high migration (due to the strong interaction between this food simulant and PS) that would indicate non-compliance (exceeding the SML). 

According to the results obtained in the present study for the HIPS sheets, only testing for 10 days at 20 °C with 10% ethanol or 20% ethanol would lead to migration levels comparable to or slightly higher than those found in refrigerated dairy products. The other tested conditions appear to be too severe and lead to higher, or significantly higher (depending on the type of simulant and the temperature), styrene migration levels into food products, such as yogurt, stored under refrigerated conditions.

According to the current legislation (Art. 18.6 of Regulation (EU) No 10/2011), verification of compliance with a SML for a material or article can be demonstrated using food in a migration test rather than a food simulant. The results of specific migration testing obtained in food prevail over the results obtained in food simulants. However, the use of foods for specific migration testing of materials and articles not yet in contact with food poses several practical constraints and analytical challenges. Therefore, migration from FCMs across European chemistry laboratories is usually tested using simulants under standardized conditions with accredited methods of analysis.

## 5. Conclusions

A precautionary specific migration limit (SML) for styrene of 40 µg/kg has been recently proposed by the EU Commission to reflect safety uncertainties and the absence of a SML [23]. 

The present study assessed the level of styrene in different refrigerated dairy products taken from the supermarket and compared these values with the results from simulated migration testing. In addition to the conventional food simulant and test conditions specified in Regulation (EU) No 10/2011 for refrigerated dairy products (50% ethanol for 10 days at 40 °C), milder simulants and test conditions (10% ethanol and 20% ethanol for 10 days at 40 °C and 20 °C) were investigated. The results obtained show that:Styrene levels in foods packed in polystyrene (refrigerated dairy products from the supermarket) at BBD ranged from 2.8 µg/kg to 22.4 µg/kg (applying the EU cube model: 6 dm^2^/kg). The higher levels were found in the single-serving portions of fresh cheese (up to approx. 22 µg/kg), while all the other investigated products packed in PS pots/cups remained below or at 10 µg/kg. All products complied with the SML of 40 µg/kg being proposed for styrene.Simulated migration testing of PS with 10%, 20%, or 50% ethanol for 10 days at 40 °C resulted in higher styrene migration levels than those found in food and exceeded the proposed SML for styrene.The use of 50% ethanol causes strong interactions with PS (weight increase > 1%) at all tested temperatures.Testing PS with 10% or 20% ethanol for 10 days at 20 °C leads to styrene migration levels that are more comparable to those found in food products stored under refrigerated conditions.

In view that a SML for styrene monomer in plastic FCMs is likely to be established in the near future, it is anticipated that the results of this study will be of value for the evaluation of the dietary exposure of the European population to styrene as well as for the selection of appropriate migration testing conditions for food packaging made of PS.

## Figures and Tables

**Table 2 foods-12-02609-t002:** Residual level in the samples and migration of styrene in food simulants at different testing conditions.

Sample	*Cp,0* [µg/g]	Testing Conditions	Food Simulant	Styrene Migration [µg/6 dm^2^] ^1^ ± SD	Styrene Relative Migration [%] ^2^	Weight Increase [%] ± SD
PS pot code-29	310 ± 5	10 days/20 °C	10% ethanol	**10.7 ± 4.5**	0.3	0.2
			20% ethanol	**12.4 ± 2**	0.3	0.3
			50% ethanol	**27.3 ± 4.7**	0.7	1.2
		10 days/40 °C	10% ethanol	**52.6 ± 5.1**	1.4	0.4
			20% ethanol	**75.6 ± 3.9**	2.0	0.7
			50% ethanol	**166.5 ± 5.2**	4.4	1.5
HIPS sheet	392 ± 8	10 days/20 °C	10% ethanol	**25.5 ± 1.4**	0.4	0.4
			20% ethanol	**35.0 ± 0.4**	0.5	0.5
			50% ethanol	**68.5 ± 0.8**	0.9	1.2
		10 days/40 °C	10% ethanol	**75.6 ± 1.1**	1.0	0.4
			20% ethanol	**113.1 ± 11.0**	1.5	0.7
			50% ethanol	**353.1 ± 13**	4.7	1.4

^1^ Calculated applying the EU cube model: 6 dm^2^/kg. ^2^ Calculated by dividing the mass of styrene recovered in the food simulant by the initial mass determined in the sample material.

## Data Availability

The data used to support the findings of this study can be made available by the corresponding author upon request.

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
