# Peer review of "Styrene Monomer Levels in Polystyrene-Packed Dairy Products from the Market versus Simulated Migration Testing"

_foods, 2023, doi:10.3390/foods12132609_

Round 1

Reviewer 1 Report

 Reviewer comments

In this manuscript, the authors claimed the “Styrene Monomer Levels in Polystyrene Packed Dairy Products from the Market versus Simulated Migration Testing’’. The manuscript needs minor revision. Bellow, find the comments:

1.      The main objective of the paper must be written on the clearer and more concise way at the end of introduction section.

2.      I suggested to authors to perform high-resolution confocal Raman spectroscope to probe the concentration profile of styrene in different refrigerated dairy 118 products taken from the supermarket.

3.      Check the English grammar in all manuscript.

4.      Conclusion is too long, please summarize it.

 Reviewer comments

In this manuscript, the authors claimed the “Styrene Monomer Levels in Polystyrene Packed Dairy Products from the Market versus Simulated Migration Testing’’. The manuscript will be accepted with minor revision. Bellow, find the comments:

1.      The main objective of the paper must be written on the clearer and more concise way at the end of introduction section.

2.      I suggested to authors to perform high-resolution confocal Raman spectroscope to probe the concentration profile of styrene in different refrigerated dairy 118 products taken from the supermarket.

3.      Check the English grammar in all manuscript.

4.      Conclusion is too long, please summarize it.

Author Response

Many thanks for this positive feedback and suggestions! In response to your observations:
1. we revised the text at the end of the introduction, explaining in a more concise way the main objective of our paper.
2. we do not have the possibility to perform such analyses because we do not have Raman spectroscopy instrumentation. However, the main aim of this study would be to compare the amount migrated to actual food products (taken from the supermarket) with the amount migrated to food simulants, therefore the applied analytical techniques are adequate to the scope.
3. a native speaker made an additional English proofreading.
4. we revised the conclusions, shortening it.

Reviewer 2 Report

The manuscript is well-written and covers a relevant and critical topic. Some comments:

- p.3, line 136: S/V (surface area to volume ratio) is usually given in L-1. Is it correct to designate "S/V" for the surface area to mass ratio (dm2/kg)?

- p.4, line 184: Please define HIPS as it first appears in the text.

- Could the product's acidity, texture, or viscosity affect the migration kinetics? And how?

- Were the thicknesses of the dairy product containers similar?

- p.5, line 204: How were test specimens dried after migration contact?

Author Response

Many thanks for this positive feedback and suggestions! In response to your observations:
- in this case it is correct to designate S/V (surface area to volume ratio) in dm2/kg (and not in L-1) assuming density of the food products = 1. The reason is that according to Regulation (EU) No 10/2011 on FCM, to check the compliance, the specific migration values shall be expressed in mg/kg.
- we added the definition of HIPS where it first appears in the text.
- we cannot exclude that the product's acidity, texture, or viscosity affect the migration kinetics. However, the main aim of this study would be to compare the amount migrated to actual food products (taken from the supermarket) with the amount migrated to food simulants, therefore we did not investigate the factors influencing migration in the different food products.
- the thicknesses of the dairy product containers were similar.
- after migration contact the test specimens were gently dried with paper towel.

Reviewer 3 Report

This manuscript presents the results of styrene levels in dairy products packaged in polystyrene in order to compare the results of a simulated migration testing. As the application of specific migration limit to styrene is under consideration, this survey will provide valuable information that will be of interest to multiple readers of Foods. However, some revisions of the manuscript will be necessary.

Comments

L118: The main aim of this study would be to compare the amount migrated to actual food products with the amount migrated to food simulants. Are there any such previous studies?

L175: More information about the validation characteristics for analytical methods is required for other authors to reproduce the analysis. Also, spiked level is not described. Typical validation characteristics for analytical methods that should be considered are: accuracy, recovery, selectivity, calibration, precision and limit of quantitation (LOQ).

L335: Same as comment above. Results of recovery tests are required to ensure data reliability.

L246: Results of dairy products packed in PS pots/cups and PS single-serving portions of fresh cheese are discussed separately in the text, but shown together in Table 1 which is confusing. Please modify Table 1, and make it clearer.

L255: Correct “18 dm2/kg” to “17 dm2/kg”.

L350-351, 391: Use consistent number of digits in the text and tables. Table 2: 10.7-25.5 ug/kg, text: 11-25 ug/kg (L350).

L401: Correct “352 ug/kg” to “353.1 ug/kg”.

Table 1: LoD level should be indicated.

Author Response

Many thanks for this positive feedback and suggestions! In response to your observations:
L118: previous studies which compare the amount migrated to actual food products with the amount migrated to food simulants are available and cited in the introduction in lines 94-98-104 and 111, the same are cited also in the discussion part.
L175: We added the validation data.
L335: Same as comment above. We added the validation data.
L246: We modified Table 1, separating the results of dairy products packed in PS pots/cups from PS single-serving portions and make it clearer.
L255: Corrected “18 dm2/kg” to “17 dm2/kg”.
L350-351, 391: revised all number of digits in the text and tables.
L401: Corrected “352 ug/kg” to “353.1 ug/kg”.
Table 1: LoD level was added.

Reviewer 4 Report

After review of the manuscript entitled “Styrene Monomer Levels in Polystyrene Packed Dairy Products from the Market versus Simulated Migration Testing”, the content explored in this article has certain scientific values, the following suggestions may be helpful in improving the quality of the manuscript:

1.       It is important to present the results of methodological investigations for quantitative analysis of styrene monomer when by using instrument, such as LOD/LOQ, precision, accuracy, et al. These are the foundation of the credibility of this study for quantification of styrene monomer in FCMs.

Author Response

Many thanks for this positive feedback and suggestions! In response to your observations:
More information about the validation characteristics for the analytical method used was required also from reviewer 3. We added the validation data in a new chapter.

Round 2

Reviewer 3 Report

The manuscript has been sufficiently improved to warrant publication in Foods. 

Reviewer 4 Report

Accepted, for consideration